# Reactive Melt Mixing of Poly(3-Hydroxybutyrate)/Rice Husk Flour Composites with Purified Biosustainably Produced Poly(3-Hydroxybutyrate-*co*-3-Hydroxyvalerate)

**DOI:** 10.3390/ma12132152

**Published:** 2019-07-04

**Authors:** Beatriz Melendez-Rodriguez, Sergio Torres-Giner, Abdulaziz Aldureid, Luis Cabedo, Jose M. Lagaron

**Affiliations:** 1Novel Materials and Nanotechnology Group, Institute of Agrochemistry and Food Technology (IATA), Spanish Council for Scientific Research (CSIC), Calle Catedrático Agustín Escardino Benlloch 7, 46980 Paterna, Spain; 2Polymers and Advanced Materials Group (PIMA), Universitat Jaume I, 12071 Castellón, Spain

**Keywords:** PHB, PHBV, rice husk, green composites, biosustainability, waste valorization

## Abstract

Novel green composites based on commercial poly(3-hydroxybutyrate) (PHB) filled with 10 wt % rice husk flour (RHF) were melt-compounded in a mini-mixer unit using triglycidyl isocyanurate (TGIC) as compatibilizer and dicumyl peroxide (DCP) as initiator. Purified poly(3-hydroxybutyrate-*co*-3-hydroxyvalerate) (PHBV) produced by mixed bacterial cultures derived from fruit pulp waste was then incorporated into the green composite in contents in the 5–50 wt % range. Films for testing were obtained thereafter by thermo-compression and characterized. Results showed that the incorporation of up to 20 wt % of biowaste derived PHBV yielded green composite films with a high contact transparency, relatively low crystallinity, high thermal stability, improved mechanical ductility, and medium barrier performance to water vapor and aroma. This study puts forth the potential use of purified biosustainably produced PHBV as a cost-effective additive to develop more affordable and waste valorized food packaging articles.

## 1. Introduction

The current concern to reduce the use of petroleum-derived materials has led to the search for natural and biodegradable polymers. Polyhydroxyalkanoates (PHAs) is a family of linear polyesters produced in nature by the action of bacteria during fermentation of sugar or lipids in famine conditions [1]. PHAs represent a good alternative to conventional polymers in the frame of the circular economy since they are fully bio-based and biodegradable [2]. Among the different commercially available PHAs, the most widely studied is poly(3-hydroxybutyrate) (PHB). This isotactic homopolyester presents a relatively high melting temperature (T_m_) and good stiffness and strength due to its high crystallinity (>50%). As a result, PHB articles present similar performance or even greater than some commodities plastics such as polypropylene (PP) and barrier properties close to those of polyethylene terephthalate (PET) [3]. PHB undergoes rapid and complete disintegration within a maximum period of 6 months through the action of enzymes and/or chemical deterioration associated with living microorganisms. Moreover, PHB is biodegradable not only in composting conditions but also in other environments such as marine water [4]. 

However, the use of PHB for packaging applications is limited due to its excessive brittleness and narrow processing temperature window [5]. To overcome these limitations, its copolymer with 3-hydroxyvalerate (HV), that is, poly(3-hydroxybutyrate-*co*-3-hydroxyvalerate) (PHBV), can result advantageous since it shows higher ductility as well as reduced crystallinity and lower T_m_ [6]. PHBV articles have been proposed to be applied in the areas of food and cosmetic packaging such as shampoo bottles, plastic beverage bottles, milk cartons, cosmetic containers, among others, due to its renewability, biodegradability, and high water vapor barrier [7,8]. However, obtaining PHAs habitually requires large investments due to both the high cost of the carbon source and the lack of efficient cultivation techniques [9]. Indeed, the current production cost of PHAs is estimated to be up to 15-fold higher than conventional polyolefins [10]. Therefore, great efforts in their industrial production are currently focused on reducing the manufacturing cost to make it more competitive [11]. In this sense, biowaste derived PHAs are both economically and environmentally attractive, in particular, those that use food waste as the raw material source. For instance, fermented cheese whey (CW), which is mostly not fully valorized at present, can be used as the feeding solution for PHA production [12]. Nowadays, most significant research efforts are targeted to optimize the extraction methods, especially in mixed cultures, and also to reduce the amounts of chemicals used to make the process environmentally friendly [13]. 

Despite of the extraordinary suitability of PHAs as candidates for sustainable food packaging applications, they are still not cost-effective due to the fermentation and downstream processes during bioreactor production [14]. In this context, a possible strategy to reduce price is the use of agro-food waste derived fillers, which also allows a more sustainable packaging concept since they valorize residues obtained from the agricultural and food industries, and thus reduce the overall impact of the industrial production cycle [15]. The combination of a bio-based and biodegradable polymer with natural fillers is habitually termed “green composite”, which means that the whole material is obtained from renewable resources and it is also biodegradable [16]. Over the past few years, the use of natural fillers to develop polymer composites has significantly increased because of their significant processing advantages, biodegradability, low cost, non-abrasive, low relative density, high specific strength, and renewable nature [17]. Moreover, these natural fillers represent an environmentally friendly solution since they decrease polluting emissions and energy requirements for processing as well as enhance energy recovery and end-of-life biodegradability [18,19,20]. In this context, different green composites have been obtained using lignocellulosic fillers derived from food, agricultural, industrial, and marine resources such as rice husk [21,22], almond husk [23,24,25], walnut shell [26], peanut shell [27], coconut fibers [28], orange peel [29], recycled cotton [30], *Posidonia oceanica* seaweed [31], etc.

Rice (*Oryza sativa* L.) is an important crop cultivated mostly in China, India, and Indonesia [32]. The annual world rice production is approximately 600 million tons, of which 20% is currently wasted as rice husk [33]. Most of this by-product is used as a bedding material for animals, burned, or landfilled, causing several environmental and health problems. Rice husk is a relatively hard material since it is typically composed of 20 wt % ash, 38 wt % cellulose, 22 wt % lignin, 18 wt % pentose, and 2 wt % of other organic components [34]. Therefore, rice husk has been used to reinforce several thermoplastics such as high-density polyethylene (HDPE) [35,36], PP [37], PP and HDPE [38], polylactide (PLA) [39], and also recently PHB [40,41,42,43]. However, the inherently poor interfacial adhesion between the lignocellulosic fillers and polymers generally yields a composite with low dispersion and a high content of particle aggregates [44]. This effect is related to the low chemical affinity between lignocellulosic fillers with most polymer and biopolymer matrices, which compromises the strength and also increases moisture absorption of the green composites [45]. To improve the interfacial adhesion between both composite components, compatibilizers or coupling agents are generally added or the filler surfaces are chemically pretreated [46]. Moreover, in the case of reactive compatibilizers, chemical bonds between the fillers and polymer matrix are formed and the overall performance of the composite can be remarkably improved [47]. For instance, the maleic anhydride (MA) grafting of PHBV matrix prior to extrusion has successfully increased the hydrophilicity of the biopolyester matrix making it more compatible with lignocellulosic fillers [48].

In this context, triglycidyl isocyanurate (TGIC) and dicumyl peroxide (DCP) can be effectively combined to compatibilize polymer composites. On the one hand, TGIC is a three-functional epoxy compound that plays a hinge-like role between lignocellulosic fillers and polyester matrices. The hydroxyl (–OH) groups of cellulose present on the fillers’ surface and the ones from the end groups of the biopolyester molecular chains, namely hydroxyl or carboxyl, can readily react with the epoxy groups of TGIC during melt blending [49]. Also, TGIC has been reported to provide a chain-extension effect on the processability of PET, increasing its molecular weight (M_W_) and potentially avoiding chain scission by hydrolysis [50]. On the other hand, DCP has been used as a free-radical grafting initiator in different polymer systems. In this sense, peroxides can form covalent carbon–carbon cross-links between the biopolymer chain segments, promoting the compatibilization of immiscible components in binary polymer blends [51] and also in polymer composites [52]. In the latter case, the addition of DCP to the composite mixture during melt mixing can give rise to both cross-linking of the polymer chains and grafting of natural fillers onto the polymer chains. Interestingly, due to the presence of three reactive –OH groups on each cellulose unit, the grafting of the cellulosic fillers onto the polymer chains dominates over the cross-linking of polymers because of the higher free radical reactivity of the –OH groups of cellulose [53]. Based on this phenomenon, different studies have for instance reported that DCP improved the mechanical properties of low-density polyethylene (LDPE)/wood fiber composites via peroxide-initiated cross-linking process [54,55,56,57].

The objective of this study is to develop highly sustainable materials with enhanced performance based on commercial PHB and rice husk fillers containing different amounts of purified PHBV that was produced by mixed bacterial cultures derived from wastes of the food industry. The green composites were prepared by melt compounding in a laboratory melt-mixer and shaped into films by thermo-compression. The resultant green composite films were characterized in terms of their morphology and optical characteristics as well as thermal, mechanical, and barrier properties in order to ascertain their potential in food packaging applications.

## 2. Materials and Methods

### 2.1. Materials

Commercial PHB homopolyester was supplied as P226F in the form of pellets by Biomer (Krailling, Germany). According to the manufacturer, this biopolymer resin presents a density of 1.25 g/cm³ and a melt flow index (MFI) of 10 g/10 min (5 kg, 180 °C). Biowaste derived PHBV copolyester was produced at pilot-plant scale at Universidade NOVA (Lisbon, Portugal) using mixed microbial cultures fed with fermented fruit pulps supplied by SumolCompal S.A. (Carnaxide, Portugal) as an industrial residue of the juice industry. The molar fraction of HV in the copolymer was ~20 mol %. The PHBV was purified with chloroform (Sigma-Aldrich S.A., Madrid, Spain) to produce a solid powder. Further details about the biopolymer and its purification route can be found elsewhere [58].

Rice husk was kindly provided by Herba Ingredients (Valencia, Spain). It was delivered in the form of flakes as a by-product of the rice industry. d-limonene, with 98% purity, TGIC (reference 379506), with a M_W_ of 297.26 g/mol, and DCP (reference 329541), with a M_W_ of 270.37 g/mol and 98% assay, were all purchased from Sigma-Aldrich S.A. (Madrid, Spain).

### 2.2. Preparation of RHF

The procedure to obtain rice husk flour (RHF) consisted on a mechanical grinding following to sieving to ensure a low particle size. For this, the native rice husk was ground in a mechanical knife mill (Thermomix TM21, Vorwerk, Madrid, Spain) and then sieved in a 140-μm mesh (TED-0300, Filtra Vibración S.L., Badalona, Spain). The resultant powder was dried at 100 °C in oven (T3060, Heraeus Instruments, Hanau, Germany) for 24 h. Figure 1 shows the as-received flakes of rice husk (Figure 1a) and the resultant RHF (Figure 1b).

### 2.3. Melt Mixing

Prior to processing, both PHA resins were dried at 60 °C for 24 h in an oven (Digitheat, JP selecta S.A., Barcelona, Spain) to remove any residual moisture. Then, different amounts of purified PHBV, from 5 wt % to 50 wt %, were manually pre-mixed with commercial PHB in a zipper bag. A fixed content of RHF was added at 10 wt % to the mixture whereas the reactive compatibilizers, that is, TGIC and DCP, were incorporated at 1 part per hundred resin (phr) and 0.25 phr of PHB/PHBV/RHF composite, respectively. A PHB/RHF composite without PHBV and a PHB/PHBV blend without RHF were also prepared as control materials. Table 1 summarizes the different formulations prepared.

To prepare the samples, a total amount of 12 g of material was melt-compounded in a 16 cm^3^ Brabender Plastograph Original E mini-mixer from Brabender GmbH & Co. KG (Duisburg, Germany). First the biopolymers and, then, the RHF powder were fed to the internal mixing chamber at a rotating speed of 60 rpm for 1 min. After this, TGIC and DCP were added and the whole composition was melt-mixed at 100 rpm for another 3 min. The processing temperature was set at 180 °C. Once the mixing process was completed, each batch was withdrawn from the mini-mixer and cooled in air to room temperature. The resultant doughs were stored in dissectors containing silica gel at 0% relative humidity (RH) and 25 °C for at least 48 h for conditioning.

The different doughs were, thereafter, thermo-compressed into films using a hot-plate hydraulic press (Carver 4122, Wabash, IN, USA). The samples were first placed in the plates at 180 °C for 1 min, without pressure, to remove any residual moisture and then hot-pressed at 4–5 bars for 3 min. Flat films with a total thickness of ~100 µm were obtained and stored in a desiccator at 25 °C and 0% RH for 15 days before characterization.

### 2.4. Characterization

#### 2.4.1. Morphology 

The morphologies of the RHF particles and the film cross-sections were observed by scanning electron microscopy (SEM) using an S-4800 device from Hitachi (Tokyo, Japan). For the cross-section observations, the films were cryo-fractured by immersion in liquid nitrogen. The samples were previously fixed to beveled holders using conductive double-sided adhesive tape and sputtered with a mixture of gold-palladium under vacuum. An accelerating voltage of 10 kV and a working distance of 15 mm were selected during SEM analysis. The estimation of the dimensions was performed by means of the ImageJ software v 1.41 (NIH, Bethesda, MD, USA) using a minimum of 20 SEM micrographs.

The particle size distribution was determined by dynamic light scattering (DLS) using a laser diffraction analyzer Mastersizer 2000 from Malvern Panalytical Ltd. (Malvern, UK). According to the manufacturer, the error for the equipment of median diameter (D_50_) is 3%. Measurements were taken under stirring to avoid settling of large particles.

#### 2.4.2. Transparency 

The light transmission of the films was determined in specimens of 50 × 30 mm^2^ by quantifying the absorption of light at wavelengths between 200 nm and 700 nm in an ultraviolet–visible (UV–vis) spectrophotometer VIS3000 from Dinko Instruments (Barcelona, Spain). The transparency (T) and opacity (O) were calculated using Equation (1) [59] and Equation (2) [60], respectively:(1)$$T\;=\;\frac{A_{600}}{L}$$
(2)$$O\;=\;A_{500}\cdot L$$
where A_500_ and A_600_ are the absorbance values at 500 nm and 600 nm, respectively, and L is the film thickness (mm).

#### 2.4.3. Color Measurements

The color of the films was determined using a chroma meter CR-400 (Konica Minolta, Tokyo, Japan). The color difference (∆E*) was calculated using the following Equation (3) [59], as defined by the Commission Internationale de l’Eclairage (CIE): ∆E* = [(∆L*)^2^ + (∆a*)^2^ + (∆b^∗^)^2^]^0.5^(3)
where ∆L*, ∆a*, and ∆b* correspond to the differences in terms of lightness from black to white, from green to red, and from blue to yellow, respectively, between the film samples and the control film of PHB/PHBV.

#### 2.4.4. Thermal Analysis

Thermal transitions were studied by differential scanning calorimetry (DSC) on a DSC-7 analyzer equipped with the cooling accessory Intracooler 2 from PerkinElmer, Inc. (Waltham, MA, USA). A two-step program under nitrogen atmosphere and with a flow rate of 20 mL/min was applied: first heating from −30 °C to 190 °C followed by cooling to −30 °C. The heating and cooling rates were both set at 10 °C/min and the typical sample weight was ~3 mg. An empty aluminum pan was used as reference whereas calibration was performed using an indium sample. The values of T_m_ and enthalpy of melting (∆H_m_) were obtained from the heating scan, while the crystallization temperature from the melt (T_c_) and enthalpy of crystallization (∆H_c_) were determined from the cooling scan. All DSC measurements were performed in triplicate.

Thermogravimetric analysis (TGA) was performed in a TG-STDA model TGA/STDA851e/LF/1600 thermobalance from Mettler-Toledo, LLC (Columbus, OH, USA). The samples, with a weight of ~15 mg, were heated from 50 °C to 800 °C at a heating rate of 10 °C/min under a flow rate of 50 mL/min of nitrogen (N_2_). All TGA measurements were also done in triplicate.

#### 2.4.5. Mechanical Tests

Tensile tests were performed on stamped dumbbell-shaped film samples sizing 115 × 16 mm^2^ using an Instron 4400 universal testing machine, equipped with a 1-kN load cell, from Instron (Norwood, MA, USA) according to the ASTM standard method D638. The tests were done using a cross-head speed of 10 mm/min. Samples were conditioned for 24 h prior to analysis and the tests were performed at room conditions, that is, 40% RH and 25 °C. A minimum of six specimens were tested for each sample.

#### 2.4.6. Permeability Tests

The gravimetric method ASTM E96-95 was used to determinate the water vapor permeability (WVP) of the films. To this end, Payne permeability cups (diameter of 3.5 cm) from Elcometer Sprl (Hermallesous-Argenteau, Belgium) were filled with 5 mL of distilled water. The films were not in direct contact with water but exposed to 100% RH on one side and secured with silicon rings. They were placed within a desiccator and sealed with dried silica gel at 0% RH and 25 °C. The control samples consisted of cups with aluminum films to estimate solvent loss through the sealing. The cups were weighted periodically using an analytical balance (±0.0001 g). WVP was calculated from the regression analysis of weight loss data vs. time, whereas the weight loss was calculated as the total loss minus the loss through the sealing. The permeability was obtained by multiplying the permeance by the film thickness. All WVP measurements were performed in triplicate.

The limonene permeability (LP), similar as described above for WVP, was measured placing 5 mL of d-limonene inside the Payne permeability cups. The cups containing the films were placed at controlled room conditions of 40% RH and 25 °C. The limonene vapor permeation rate (LPRT) values were estimated from the steady-state permeation slopes and the weight loss was calculated as the total cell loss minus the loss through the sealing. LP was calculated taking into account the average film thickness in each case. LP measurements were performed in triplicate.

#### 2.4.7. Statistical Analysis

The optical, thermal, mechanical, and barrier properties were evaluated through analysis of variance (ANOVA) using STATGRAPHICS Centurion XVI v 16.1.03 from StatPoint Technologies, Inc. (Warrenton, VA, USA). Fisher’s least significant difference (LSD) was used at the 95% confidence level (*p* < 0.05). Mean values and standard deviations were also reported.

## 3. Results

### 3.1. Morphology of RHF Particles

The morphology of the RHF powder was observed by SEM for determining the particle size and shape. In the low-magnification SEM image, shown in Figure 2a, one can see that the particles were not uniform in morphology and their dimension varied broadly. In particular, small particles slightly below 10 µm, in the form of rod-like particles or rectangular junks, coexisted with short fibers, having lengths above 100 µm. The magnified SEM image, included in Figure 2b, illustrates that the outer surface of rice husk was relatively smooth but its inner part was densely covered with orderly bulges. In this regard, it has been reported that the globular structure of rice husk is responsible for its high absorption capacity [61], which can also positively contribute to favoring mechanical interactions with the biopolymer matrix. Similar morphologies were reported for instance by Schneider et al. [62], in which the RHF particles also presented a heterogeneous morphology, namely larger particles of different textures, some smoother, others rough, with also grooves along the structure. The histogram of particle size of RHF is shown in Figure 2c, where one can see that the average fiber length was ~190 µm whereas the diameter corresponding to 90% cumulative (D_90_) was ~320 µm.

### 3.2. Morphology of PHB/PHBV/RHF Films

The morphology of the film cross-sections was observed by SEM and the images are gathered in Figure 3. One can observe that all films presented a smooth and featureless fracture surface without much deformation, corresponding to a typical brittle fracture behavior. The presence of the RHF particles can be observed in the cross-sections of all the composite film samples, that is, Figure 3a,c–g, whereas the surface of the PHB/PHBV film in Figure 3b suggests that the matrix was monophasic. Interestingly, the images taken at higher magnifications revealed that RHF fillers were tightly bonded to the polymer as inferred from the absence of gap between the particles and the biopolymer matrix. Moreover, no evidence of filler pull-out or void formation was noticed. At the same time, a rough surface attributed to the matrix deformation can be observed in the fillers surrounding. Additionally, RHF aggregates were not observed but individual particles appeared regularly distributed along the biopolymer matrix indicating that an effective mixing was attained.

The above observation suggests that a high interfacial adhesion was achieved in the composites due to the combined use of TGIC and DCP during melt processing. Similar morphologies were also described, for instance, by Rosa et al. [63] for PP/RHF composites when MA-modified polypropylene (MAPP) was added as coupling agent. The presence of MAPP successfully reduced the voids sizes and turned the surface more homogeneous, confirming its effect on promoting adhesion in the interfacial region. In relation to TGIC, Hao and Wu [49] recently showed that the addition during melt blending of the isocyanurate additive improved the interfacial adhesion of PLA/sisal fibers (SF) composites. The compatibilization achieved in the green composite was ascribed to the reaction of the –OH groups present at end groups of the biopolyester molecular chains and on the cellulose surface with the epoxy functional groups of TGIC. In the first reaction, ester bonds are known to be formed with the PHA chains by glycidyl esterification of carboxylic acid end groups, which precedes hydroxyl end group etherification [64]. The second reaction generates C–O–C bonds with subsequent hydroxyl side-group formation on the cellulose surface [29]. This reactive compatibilization habitually leads to green composites with enhanced performance properties [65]. For DCP, Wei et al. [52,66] recently described the coupling mechanism of cellulose to PHB and PHBV. Briefly, when the peroxide is exposed to heat during extrusion it decomposes into strong free radicals, which tend to abstract hydrogens (H’) from the biopolymer and cellulose molecular chains and initiate the grafting process between the two phases in the composites. The authors postulated that the grafted copolymers formed on the interfaces of cellulose and PHA coupled the hydrophilic filler to the hydrophobic biopolymer matrix. Therefore, grafting of RHF onto the PHB/PHBV matrix was successfully achieved by the formation of low concentrations of DCP derived radicals at high temperature during extrusion that initiated both the formation both H abstraction and triggered the reaction of the epoxy groups of TGIC with the OH groups of both cellulose and the terminal groups of the biopolyester chains. In relation to the addition of the biowaste derived PHBV, a good mixture between the two PHAs was attained since it was not possible perceive the presence of two phases in the biopolymer matrix, even at the highest PHBV content, that is, 50 wt %. In the film surfaces one can still observe the presence of some remaining impurities, which may be ascribed to organic remnants of small amounts of cell debris or fatty acids from the bioproduction process of PHBV. A similar morphology was recently reported by Martinez-Abad et al. [67] for PHB/unpurified PHBV blends, who also observed a good degree of interaction between the commercial homopolyester and the biosustainably produced copolyester. 

### 3.3. Optical Properties of PHB/PHBV/RHF Films

The visual aspect of the films is displayed in Figure 4 to ascertain their optical properties. Simple naked eye examination of the films indicates that all the samples were slightly opaque but also showed high contact transparency. Additionally, the composite films developed a yellow-to-brown color due to the intrinsic natural color of the RHF powder, which one can observe in previous Figure 1b.

To quantify the color change resulted from the addition of PHBV in PHB/RHF, the color coordinates (a*, b*, L*) and the values of ∆E*, T, and O were determined and reported in Table 2. One can observe that the a* b* coordinates of the PHB/RHF film confirmed the above observed yellow-to-brown color of the sample whereas the PHB/PHBV film presented a natural color. The incorporation of PHBV into the green composite films resulted in a slightly increased the value of a* and, most notably, of b*, confirming the development of a brown tonality. Moreover, the composite films became darker since the L* values were also reduced. In any case, the color differences in the composite films containing different amounts of PHBV were relatively low, that is, ∆E* values remained below 6. One can also observe that the composite films also presented slightly lower transparency and higher opacity with the increasing PHBV contents, having values of T and O around 10–15% and 5% higher, respectively. This haze increase is typically observed in polymer blends due to differences in the refractive index of light diffraction of each polymer [68]. Although the brownish color of the film samples may seem a disadvantage for food packaging, it also offers the capacity to partially block the transmission of ultraviolet and visible (UV–vis) light. This can be a desirable attribute since the films can provide more protection to foodstuff from light, especially UV radiation, which can cause lipid oxidation in the food products [60,69].

### 3.4. Crystallinity of PHB/PHBV/RHF Films

Figure 5 displays the DSC thermograms of the film samples obtained during the heating (Figure 5a) and cooling (Figure 5b) scans. Table 3 displays the main thermal transitions obtained from the DSC curves. One can observe that melting in the PHB/RHF film took place in two peaks, which were noted as T_m1_ and T_m2_, whereas the unfilled PHB/PHBV film melted in a single peak. The double-melting peak phenomenon can be ascribed to the formation of crystalline structures with dissimilar lamellae thicknesses or the presence of crystallite blocks with different degrees of perfection [30]. The first peak originates from the melting of the PHB fraction that crystallized previously during the film formation, while in the second peak contributes the melting of the recrystallized PHB fraction during heating. In this context, other works have reported that melt-extruded films of neat PHB melt in 170–175 °C range [70,71]. Then, one can consider that the presence of the RHF fillers restricted the chain-folding process of PHB during crystallization.

Interestingly, the incorporation into PHB/RHF of low contents of PHBV, that is, 5 wt %, yielded the film sample with the highest melting peak, that is, ~172 °C, whereas it also suppressed the double-melting peak behavior. This observation points out that the co-addition of 5 wt % PHBV and 10 wt % RHF enhanced the crystallization of PHB molecules, which was further supported by the shift of T_c_ from 113.1 °C, for the PHB/RHF film, to 118.9 °C, for the PHB/PHBV5/RHF film. The addition of higher contents of PHBV, however, led to films with two melting peaks. The first melting peak can be related to the PHBV-rich fraction, which was seen in the 158–166 °C range, whereas the second one corresponds to PHB melting, observed in 168–174 °C range. This result points out that fully miscibility was only attained in the composite blends containing low amounts of PHBV, that is, 5–10 wt %. At higher PHBV contents, the composite films formed a two-phase system and the T_m_ values were also reduced. Thus, the films produced with PHBV contents higher than 10 wt % showed a two phase crystallization, having a liquid–liquid separation at temperatures higher than 165 °C. One assumes that in these samples, due to the fact that the HV content in PHBV was relatively high, that is, 20 mol %, phase segregation preceded co-crystallization. One can also observe that the melting enthalpy of the first peak decreased while that of the second peak increased with the increase of the PHBV content in the blend. This observation indicates that crystallization occurred mainly in the PHB-rich regions by which the HV units were partially included into the PHB lattice and also induced some defects in the biopolymer crystals [72]. In this regards, different studies on the crystallization behavior of PHB/PHBV blends have indicated that their degree of miscibility decreases gradually as the HV in the copolyester increases. For instance, blends of PHB with PHBV were fully miscible over up to approximately 60 wt % of copolyester with a HV content of 18.4 mol % [73]. On the contrary, blends consisting of PHB and PHBV of high contents of HV, that is, 76 mol %, showed no depression of the melting point of each PHA, indicating total immiscibility [74]. In another work, Saito et al. [75] studied the competition between co-crystallization and phase segregation in blends of PHB and PHBV with different HV contents. The authors observed almost perfect co-crystallization in blends based on PHB and PHBV with 9 mol % HV, whereas HV contents >15 mol % induced phase segregation, that is, increased the percentage of PHBV that segregates from the growth front of crystals prior to co-crystallization.

### 3.5. Thermal Stability of PHB/PHBV/RHF Films

TGA curves are plotted in Figure 6 and the most relevant properties obtained from the curves are listed in Table 4. In relation to RHF, one can observe three main weight losses. The first one occurred around 100 °C, showing a mass loss of ~2%, which corresponds to the moisture released from the lignocellulosic filler. Following the TGA curve of RHF, the second and main degradation peak started at approximately 180 °C and ended at 340 °C with an average mass loss of ~47%. This weight loss includes the degradation of low-M_W_ components, mainly hemicellulose, followed by cellulose degradation. This zone represents the main devolatilization step of biomass pyrolysis and it is referred as the “active pyrolysis zone” since mass loss rate is high [76]. After this, RHF gradually degraded over a range of temperature from approximately 340 °C to 650 °C with a mass loss of ~16%, which can be seen as a tailing in both TGA and DTG curves. This mass loss is related to the degradation of lignin and it is called “passive pyrolysis zone” since the percentage of mass loss is smaller and the mass loss rate is also much lower compared to that in the second zone [76]. Indeed, when the temperature reached 650 °C, the degradation rates were no longer significant as most volatiles were already pyrolyzed. The rest was converted into char and gases, resulting in a residual mass of nearly 33%, which can be related to the high silicate content in rice husk [47].

Thermal degradation of the unfilled PHB/PHBV film occurred through a single and sharp degradation step that ranged from about 250 °C to 290 °C. PHA degradation typically follows a random chain scission model of the ester linkage that involves a *cis*-elimination reaction of β-CH and a six-member ring transition to form crotonic acid and its oligomers [77]. The presence of RHF reduced both the onset degradation temperature (T_5%_) and degradation temperature (T_deg_) by approximately 10 °C and 15 °C, respectively, and also generated a low-intensity second mass centered around 330 °C. This reduction in thermal stability can be mainly ascribed to the lower degradation temperature of the lignocellulosic particles as well as to the presence of some remaining water. One can also observe that all composite films presented a similar thermal degradation profile though certain stability increase was attained for the highest PHBV contents. For instance, the T_5%_ value increased from 242.5 °C, for the PHB/RHF film, to 257.2 °C, for the PHB/PHBV50/RHF film. Similarly, the values of T_deg_ also increased from 270 °C, for the PHB/RHF film, to 275.5 °C, for the PHB/PHBV50/RHF film. The improvement achieved in the thermal stability can be ascribed to the high purity of the PHBV incorporated in the blend, based on the previous selection of the optimal purification route [58], which has also been reported to be more thermally stable than PHB [78]. Finally, the amount of residual mass was in the 3–4% range, being mainly ascribed to the formed char from RHF.

### 3.6. Mechanical Properties of PHB/PHBV/RHF Films

Figure 7 shows the tensile stress–strain curves, obtained at room temperature, for the thermo-compressed films. The mechanical results, in terms of tensile modulus (E), tensile strength at yield (σ_y_), elongation at break (ε_b_), and toughness are summarized in Table 5. One can see that all the films were relatively stiff and also brittle due to the intrinsically high crystallinity of PHB, which is derived from secondary crystallization that occurs post-processing with age [79]. This mechanical brittleness habitually represents an obstacle to the practical applications of PHB, for instance in packaging. The presence of RHF further promoted rigidity and brittleness of PHB, showing *E* and ε_b_ values of 3025 MPa and 1.1%, respectively. One can also observe that the unfilled PHB/PHBV film showed the lowest E value, that is, 1785 MPa, and also the highest values of ε_b_ and toughness, that is, 8.4% and 1.9 mJ/m^3^, respectively. Whereas the mechanical properties of the PHB/RHF and PHB/PHBV5/RHF films were relatively similar, the incorporation of 10 wt % PHBV into PHB/RHF successfully resulted in a film with very balanced performance in terms of mechanical strength and ductility. In particular, the PHB/PHBV10/RHF film showed moderate values of *E* and σ_y_, that is, 2508 MPa and 30.4 MPa, respectively, and a 3-fold increase in ε_b_ and remarkable higher toughness in comparison to the PHB/RHF film. However, the films produced at higher PHBV contents, that is, 20–50 wt %, showed lower and relatively similar performance. These results indicate that PHBV can successfully produce a toughening effect on the PHB matrix at low or intermediate contents, which can be ascribed to the high solubility achieved in these films samples as described above during the crystallinity analysis. The resultant lower stiffness and higher flexibility of these PHBV-containing films can be attributed to the presence of dislocations, crystal strain, and smaller crystallites in the PHB/PHBV soluble regions due to the insertion of the HV units into the PHB lattice, which acted as defects in the HB crystals [80]. 

### 3.7. Barrier Performance of PHB/PHBV/RHF Films

Table 6 finally gathers the WVP and LP values of the PHB-based films. The barrier performance to water vapor is one of the main parameters of application interest in packaging, while d-limonene is a commonly used standard compound to test mass transport of aromas. The PHB/RHF film showed somewhat higher WVP values than compression-molded PHB films previously reported by our group, that is, 1.7–1.75 × 10^−15^ kg·m/s·m^2^·Pa [81,82]. The lower water vapor barrier attained for the here-prepared green composite films can be related to the high tendency of the lignocellulosic fillers to adsorb water since transport of water vapor molecules is mainly a diffusivity-driven property in PHAs due to their intrinsically low hydrophilic character [83]. The incorporation of biowaste derived PHBV into PHB/RHF increased slightly the WVP values in the films. At low PHBV contents, however, the barrier performance of the composite films was relatively similar since the samples showed variations close to the detection limit of the technique, with deviations oscillating in the 6.0–7.5 × 10^−16^ kg·m/s·m^2^·Pa range. Only the films with the highest PHBV contents, that is, 30 wt % and 50 wt %, displayed a significant decrease in the water barrier properties, which was still within the same order of magnitude. The barrier drop observed can be attributed to the higher contribution of the biomass derived PHBV, which results in an overall decrease in molecular order and crystallinity, the presence of defects and discontinuities across the polymer morphology as well as certain plasticization induced by the remaining biomass impurities of PHBV with consequent increase in free volume [67].

Similar to WVP, one can observe that the incorporation of PHBV tended to decrease the barrier properties to d-limonene of the PHB/RHF composite films. This aroma compound, as opposed to moisture, is a strong plasticizing component for PHAs, thus, solubility plays a more important role in permeability than diffusion. The LP increase observed suggests that the presence of PHBV favored an increased sorption of d-limonene in the film. Indeed, our previous studies dealing with PHA materials have shown that the LP value of films made of PHBV with 12 mol % HV is two order of magnitude higher, that is, 1.99 × 10^−13^ kg·m/s·m^2^·Pa [84], than that of neat PHB films, that is, 1.95 × 10^−15^ kg·m/s·m^2^·Pa [82]. In any case, the present results indicate that the incorporation of up to 20 wt % PHBV does not significantly affect the barrier properties of the PHB-based films against water or aroma. In a more packaging oriented application context, the composite films containing low amounts of PHBV present WVP values in the same order of magnitude than those films of petroleum-based PET, that is, 2.30 × 10^−15^ kg·m/s·m^2^·Pa [85], which is typically used in medium-barrier applications. In terms of d-limonene, the here-prepared PHB/PHBV/RHF films are two order of magnitude more barrier than compression-molded PET films, that is, 1.17 × 10^−13^ kg·m/s·m^2^·Pa [81].

## 4. Discussion

Results showed an optimum morphology with a regular distribution of RHF, tightly bonded and with absence of gap along the PHB matrix due to the use of reactive compatibilizers. The biowaste derived PHBV also showed a good miscibility with the PHB/RHF composite system, in particular at the lowest contents, that is, 5–10 wt %. At higher concentrations, however, a two-phases system was attained, indicating that crystallization occurred mainly in the PHB-rich regions. The incorporation of PHBV also increased the thermal stability of PHB/RHF, increasing the processing window of the films. With respect to the mechanical properties, contents in the 5–10 wt % range of PHBV yielded films with a more balanced performance in terms of strength and ductility, counteracting the stiffness and fragility induced by RHF. Finally, although the incorporation of PHBV increased the permeability of the films, the water vapor barrier properties of the PHB/PHBV/RHF films still remained in values close to those of PET films, whereas they still presented a high barrier to aroma.

## 5. Conclusions

The use of natural fillers and biopolymers obtained from agro-food waste currently represents a sustainable alternative to petroleum-based materials. The green composite films prepared herein are potential candidates to be used in rigid packaging for low and medium barrier applications, being processable by current conventional machinery. The valorization of agro-food waste, as well as the relative preservation of physicochemical properties, supports the use of purified biosustainably produced PHBV in the food packaging industry to develop more cost-effective PHA-based articles.

## Figures and Tables

**Figure 1 materials-12-02152-f001:**
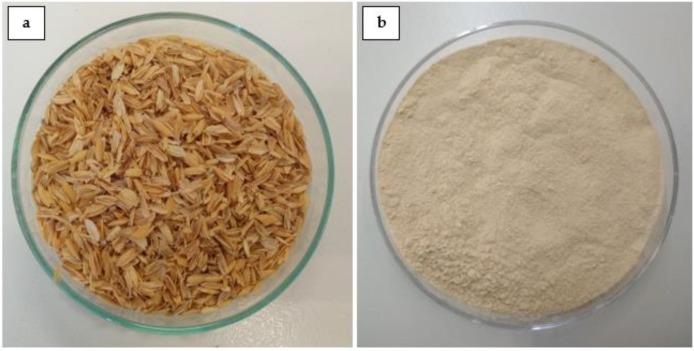
(**a**) As-received flakes of rice husk; (**b**) rice husk flour (RHF).

**Figure 2 materials-12-02152-f002:**
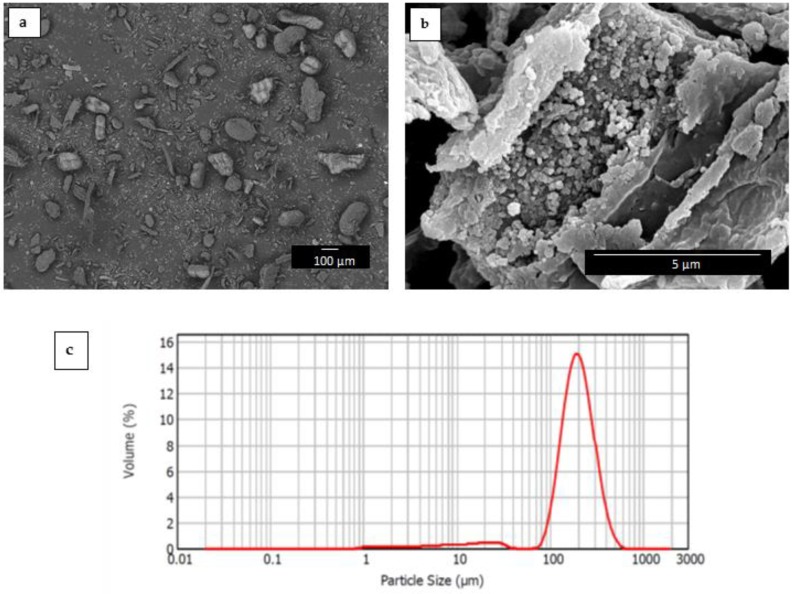
Scanning electron microscopy (SEM) images of rice husk flour (RHF) taken at (**a**) 50× with scale marker of 100 µm and (**b**) 10,000× with scale marker of 5 µm; (**c**) particle size histogram of RHF.

**Figure 3 materials-12-02152-f003:**
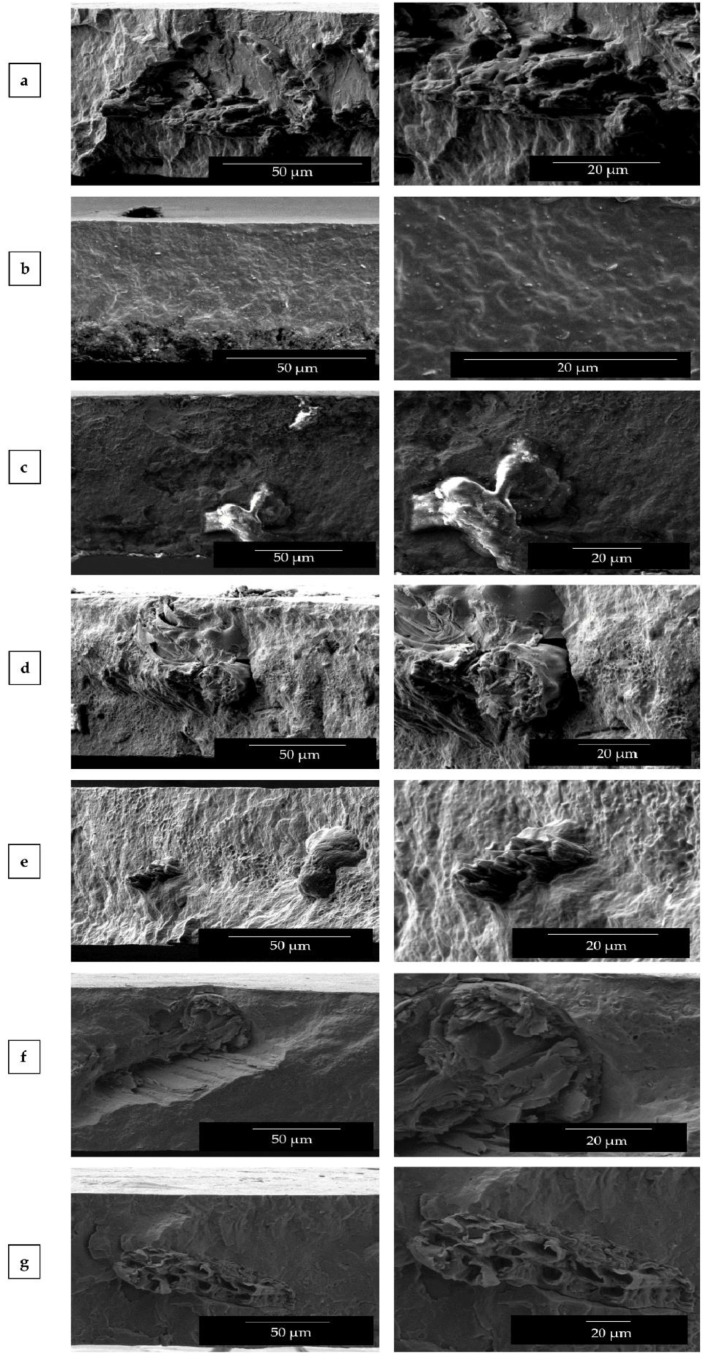
Scanning electron microscopy (SEM) images of the cross-sections of the thermo-compressed films made of poly(3-hydroxybutyrate) (PHB), poly(3-hydroxybutyrate-*co*-3-hydroxyvalerate) (PHBV), and rice husk flour (RHF): (**a**) PHB/RHF; (**b**) PHB/PHBV; (**c**) PHB/PHBV5/RHF; (**d**) PHB/PHBV10/RHF; (**e**) PHB/PHBV20/RHF; (**f**) PHB/PHBV30/RHF; (**g**) PHB/PHBV50/RHF. Images were taken at 1000× with scale markers of 50 µm (left column) and at 2500× with scale markers of 20 µm (right column).

**Figure 4 materials-12-02152-f004:**
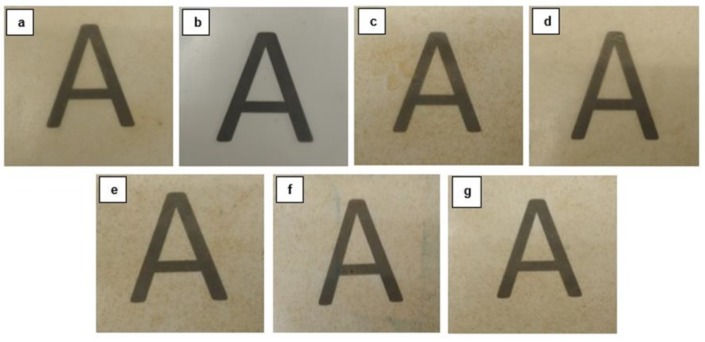
Visual aspect of the thermo-compressed films made of poly(3-hydroxybutyrate) (PHB), poly(3-hydroxybutyrate-*co*-3-hydroxyvalerate) (PHBV), and rice husk flour (RHF): (**a**) PHB/RHF; (**b**) PHB/PHBV; (**c**) PHB/PHBV5/RHF; (**d**) PHB/PHBV10/RHF; (**e**) PHB/PHBV20/RHF; (**f**) PHB/PHBV30/RHF; (**g**) PHB/PHBV50/RHF.

**Figure 5 materials-12-02152-f005:**
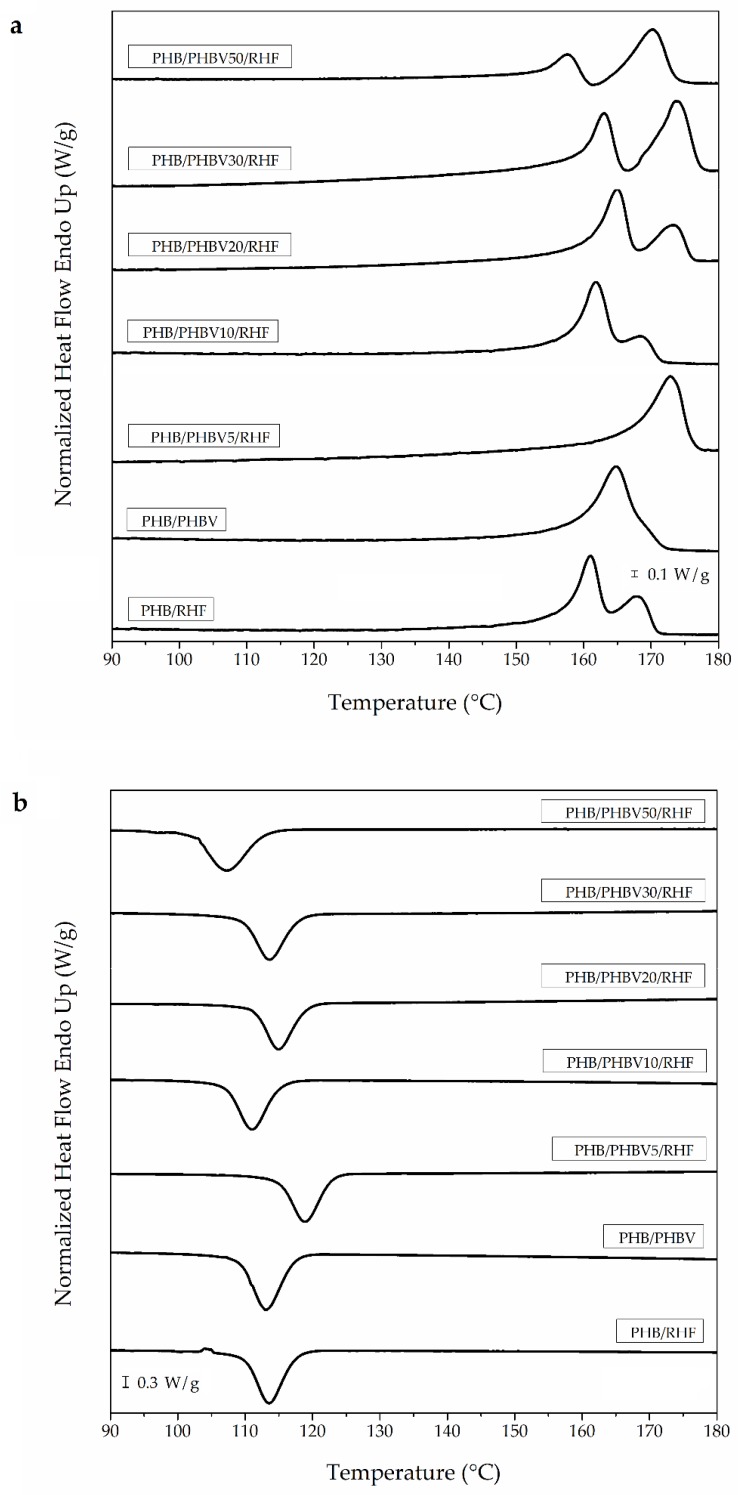
Differential scanning calorimetry (DSC) curves during (**a**) heating and (**b**) cooling of the thermo-compressed films made of poly(3-hydroxybutyrate) (PHB), poly(3-hydroxybutyrate-*co*-3-hydroxyvalerate) (PHBV), and rice husk flour (RHF).

**Figure 6 materials-12-02152-f006:**
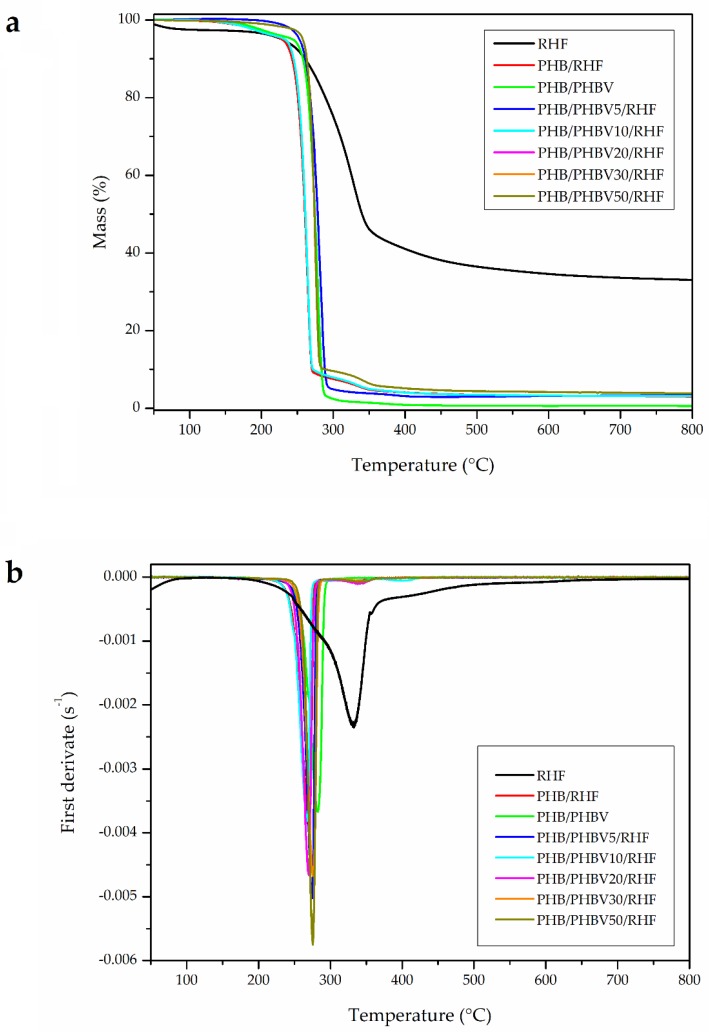
(**a**) Thermogravimetric analysis (TGA) and (**b**) first derivative (DTG) curves of the thermo-compressed films made of poly(3-hydroxybutyrate) (PHB), poly(3-hydroxybutyrate-*co*-3-hydroxyvalerate) (PHBV), and rice husk flour (RHF).

**Figure 7 materials-12-02152-f007:**
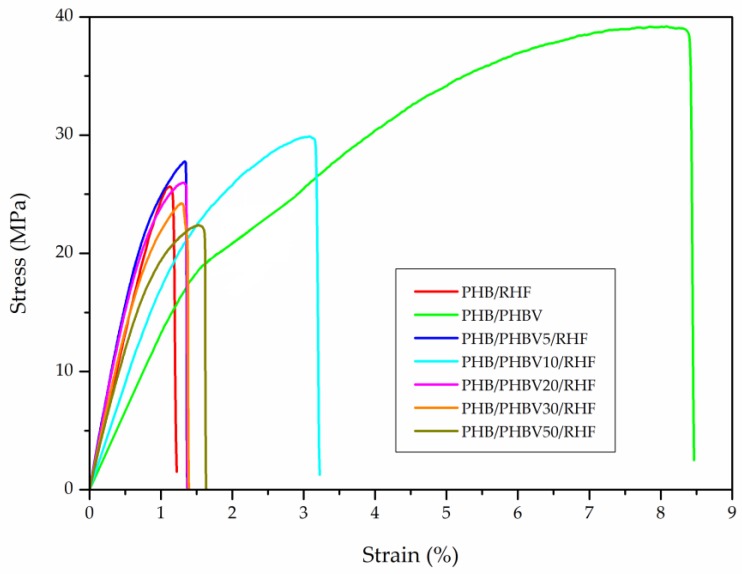
Tensile stress–strain curves of the thermo-compressed films made of poly(3-hydroxybutyrate) (PHB), poly(3-hydroxybutyrate-*co*-3-hydroxyvalerate) (PHBV), and rice husk flour (RHF).

**Table 1 materials-12-02152-t001:** Set of formulations prepared according to the weight content (wt %) of poly(3-hydroxybutyrate) (PHB), poly(3-hydroxybutyrate-*co*-3-hydroxyvalerate) (PHBV), and rice husk flour (RHF) in which triglycidyl isocyanurate (TGIC) and dicumyl peroxide (DCP) were added as parts per hundred resin (phr) of PHB/PHBV/RHF composite.

Sample	PHB (wt %)	PHBV (wt %)	RHF (wt %)	TGIC (phr)	DCP (phr)
PHB/RHF	90	0	10	1	0.25
PHB/PHBV	90	10	0	1	0.25
PHB/PHBV5/RHF	85	5	10	1	0.25
PHB/PHBV10/RHF	80	10	10	1	0.25
PHB/PHBV20/RHF	70	20	10	1	0.25
PHB/PHBV30/RHF	60	30	10	1	0.25
PHB/PHBV50/RHF	40	50	10	1	0.25

**Table 2 materials-12-02152-t002:** Optical properties of the thermo-compressed films made of poly(3-hydroxybutyrate) (PHB), poly(3-hydroxybutyrate-*co*-3-hydroxyvalerate) (PHBV), and rice husk flour (RHF).

Film	a*	b*	L*	∆E*	T	O
PHB/RHF	−0.04 ± 0.01 ^a^	11.82 ± 0.03 ^a^	80.20 ± 0.02 ^a^	-	9.08 ± 0.07 ^a^	0.107 ± 0.03 ^a^
PHB/PHBV	−0.25 ± 0.03 ^b^	1.94 ± 0.02 ^b^	86.40 ± 0.04 ^b^	-	8.48 ± 0.03 ^b^	0.099 ± 0.02 ^a^
PHB/PHBV5/RHF	0.44 ± 0.05 ^c^	14.60 ± 0.02 ^c^	78.05 ± 0.03 ^c^	3.55 ± 0.05 ^a^	9.06 ± 0.05 ^a^	0.111 ± 0.03 ^a^
PHB/PHBV10/RHF	0.52 ± 0.04 ^c^	14.84 ± 0.07 ^c^	77.62 ± 0.03 ^d^	4.01 ± 0.06 ^b^	9.91 ± 0.04 ^c^	0.116 ± 0.05 ^a^
PHB/PHBV20/RHF	0.56 ± 0.05 ^c^	15.28 ± 0.06 ^d^	77.56 ± 0.05 ^d^	4.39 ± 0.04 ^c^	10.05 ± 0.03 ^d^	0.116 ± 0.02 ^a^
PHB/PHBV30/RHF	0.70 ± 0.07 ^d^	15.36 ± 0.05 ^d^	77.29 ± 0.06 ^e^	4.64 ± 0.03 ^d^	10.18 ± 0.08 ^e^	0.117 ± 0.04 ^a^
PHB/PHBV50/RHF	1.12 ± 0.08 ^e^	16.17 ± 0.03 ^e^	76.36 ± 0.04 ^f^	5.92 ± 0.06 ^e^	10.17 ± 0.05 ^e^	0.113 ± 0.04 ^a^

a*: red/green coordinates (+a red, −a green); b*: yellow/blue coordinates (+b yellow, −b blue); L*: Luminosity (+L luminous, −L dark); ∆E*: color differences; T: transparency; O: opacity. ^a–f^ Different letters in the same column indicate a significant difference among the samples (*p* < 0.05).

**Table 3 materials-12-02152-t003:** Main thermal parameters of the thermo-compressed films made of poly(3-hydroxybutyrate) (PHB), poly(3-hydroxybutyrate-*co*-3-hydroxyvalerate) (PHBV), and rice husk flour (RHF) in terms of: crystallization temperature (T_c_), normalized enthalpy of crystallization (∆H_c_), melting temperature (T_m_), and normalized melting enthalpy (∆H_m_).

Film	T_c_ (°C)	∆H_c_ (J/g)	T_m1_ (°C)	T_m2_ (°C)	∆H_m_ (J/g)
PHB/RHF	113.1 ± 0.6 ^a^	70.2 ± 0.2 ^a^	160.5 ± 0.8 ^a^	167.2 ± 0.9 ^a^	71.8 ± 2.3 ^a^
PHB/PHBV	113.3 ± 0.3 ^a^	73.3 ± 0.4 ^b^	165.1 ± 0.3 ^b^	-	76.9 ± 1.4 ^b^
PHB/PHBV5/RHF	118.9 ± 0.6 ^b^	59.8 ± 1.8 ^c^	171.9 ± 1.5 ^c^	-	63.7 ± 4.8 ^c^
PHB/PHBV10/RHF	110.9 ± 0.2 ^c^	60.1 ± 3.9 ^c^	162.6 ± 1.1 ^d^	168.3 ± 0.4 ^a^	63.4 ± 1.6 ^c^
PHB/PHBV20/RHF	115.4 ± 0.6 ^d^	78.0 ± 4.1 ^d^	166.0 ± 1.5 ^b^	173.3 ± 0.3 ^b^	61.4 ± 0.3 ^c^
PHB/PHBV30/RHF	113.2 ± 0.6 ^a^	72.7 ± 4.7 ^d^	162.6 ± 0.7 ^d^	173.9 ± 0.1 ^c^	59.4 ± 2.2 ^c^
PHB/PHBV50/RHF	107.7 ± 0.6 ^e^	64.8 ± 0.8 ^c^	158.4 ± 1.1 ^e^	171.0 ± 0.8 ^d^	53.8 ± 0.1 ^d^

^a–e^ Different letters in the same column indicate a significant difference among the samples (*p* < 0.05).

**Table 4 materials-12-02152-t004:** Main thermal parameters of the thermo-compressed films made of poly(3-hydroxybutyrate) (PHB), poly(3-hydroxybutyrate-*co*-3-hydroxyvalerate) (PHBV), and rice husk flour (RHF) in terms of: onset temperature of degradation (T_5%_), degradation temperature (T_deg_), mass loss at T_deg_, and residual mass at 800 °C.

Film	T_5%_ (°C)	T_deg_ (°C)	Mass Loss (%)	Residual Mass (%)
RHF	228.9 ± 1.5 ^a^	335.2 ± 0.7 ^a^	28.5 ± 0.8 ^a^	33.1 ± 0.3 ^a^
PHB/RHF	242.5 ± 1.2 ^b^	270.0 ± 0.5 ^b^	67.4 ± 0.7 ^b^	3.1 ± 1.3 ^b^
PHB/PHBV	252.6 ± 1.9 ^c^	282.8 ± 0.4 ^c^	69.4 ± 1.5 ^b^	0.6 ± 0.1 ^c^
PHB/PHBV5/RHF	242.5 ± 1.5 ^b^	268.2 ± 0.5 ^d^	65.4 ± 0.3 ^c^	3.5 ± 0.2 ^b^
PHB/PHBV10/RHF	247.1 ± 1.3 ^d^	270.0 ± 0.5 ^b^	63.9 ± 0.4 ^d^	3.1 ± 0.8 ^b^
PHB/PHBV20/RHF	247.5 ± 1.4 ^d^	274.6 ± 0.6 ^e^	69.8 ± 1.2 ^b^	3.3 ± 0.9 ^b^
PHB/PHBV30/RHF	253.5 ± 1.7 ^c^	275.5 ± 0.6 ^e^	61.7 ± 0.2 ^e^	3.2 ± 0.2 ^b^
PHB/PHBV50/RHF	257.2 ± 1.6 ^e^	275.5 ± 0.6 ^e^	61.5 ± 0.3 ^e^	3.8 ± 0.4 ^b^

^a–e^ Different letters in the same column indicate a significant difference among the samples (*p* < 0.05).

**Table 5 materials-12-02152-t005:** Mechanical properties of the thermo-compressed films made of poly(3-hydroxybutyrate) (PHB), poly(3-hydroxybutyrate-*co*-3-hydroxyvalerate) (PHBV), and rice husk flour (RHF) in terms of: tensile modulus (E), tensile strength at yield (σ_y_), elongation at break (ε_b_), and toughness.

Film	E (MPa)	σ_y_ (MPa)	ε_b_ (%)	Toughness (mJ/m^3^)
PHB *	~2900 ^a^	~37 ^a^	~4 ^a^	-
PHB/RHF	3025 ± 101 ^b^	26.7 ± 2.7 ^b^	1.1 ± 0.1 ^b^	0.1 ± 0.1 ^a^
PHB/PHBV	1785 ± 129 ^c^	38.9 ± 2.0 ^a^	8.4 ± 1.1 ^c^	1.9 ± 0.3 ^b^
PHB/PHBV5/RHF	2985 ± 119 ^a,b^	27.2 ± 1.1 ^b^	1.4 ± 0.1 ^d^	0.3 ± 0.1 ^a^
PHB/PHBV10/RHF	2508 ± 207 ^d^	30.4 ± 2.7 ^b^	3.3 ± 1.3 ^a^	0.8 ± 0.3 ^c^
PHB/PHBV20/RHF	2830 ± 193 ^a,b,d^	26.5 ± 2.5 ^b^	1.2 ± 0.3 ^b,d^	0.1 ± 0.1 ^a^
PHB/PHBV30/RHF	2765 ± 201 ^a,b,d^	23.9 ± 0.7 ^b^	1.3 ± 0.2 ^b,d^	0.2 ± 0.1 ^a^
PHB/PHBV50/RHF	2649 ± 104 ^d^	19.5 ± 3.0 ^c^	1.3 ± 0.4 ^b,d^	0.2 ± 0.1 ^a^

* Average mechanical properties for compression-molded films of neat PHB [79]. ^a–d^ Different letters in the same column indicate a significant difference among the samples (*p* < 0.05).

**Table 6 materials-12-02152-t006:** Water vapor permeability (WVP) and d-limonene permeability (LP) of the thermo-compressed films made of poly(3-hydroxybutyrate) (PHB), poly(3-hydroxybutyrate-*co*-3-hydroxyvalerate) (PHBV), and rice husk flour (RHF).

Film	WVP × 10^15^ (kg·m/m^2^·Pa·s)	LP × 10^15^ (kg·m/m^2^·Pa·s)
PHB *	1.75 ^a^	1.95 ^a^
PHB/RHF	4.52 ± 0.38 ^b^	2.58 ± 1.35 ^a^
PHB/PHBV	3.27 ± 0.15 ^c^	2.04 ± 0.19 ^a^
PHB/PHBV5/RHF	4.55 ± 0.75 ^b^	2.16 ± 0.54 ^a^
PHB/PHBV10/RHF	5.03 ± 0.64 ^b^	3.10 ± 0.65 ^a^
PHB/PHBV20/RHF	5.36 ± 0.69 ^b^	3.38 ± 0.63 ^a^
PHB/PHBV30/RHF	6.01 ± 0.60 ^b^	3.72 ± 0.32 ^a^
PHB/PHBV50/RHF	7.46 ± 0.90 ^b^	5.04 ± 1.50 ^a^

* Barrier data for compression-molded PHB films reported in literature [82]. ^a–c^ Different letters in the same column indicate a significant difference among the samples (*p* < 0.05).

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
