# Peer review of "Reactive Melt Mixing of Poly(3-Hydroxybutyrate)/Rice Husk Flour Composites with Purified Biosustainably Produced Poly(3-Hydroxybutyrate-co-3-Hydroxyvalerate)"

_materials, 2019, doi:10.3390/ma12132152_

Reviewer 1 Report

The paper by Lagaron et al. reports on the preparation and characterization of new biodegradable poly(3-hydroxybutyrate) (PHB)/rice husk flour (RHF) composites; the optical transparency, crystallinity, thermal stability, mechanical performance, barrier properties and morphological behavior of the obtained composites are assessed and correlated with the structure and composition of the composites. Overall, this paper is well characterized and the data is well presented. While I am unqualified to comment on the bio-sustainable technologies of this work, I found this work to be attractive for improving the physical performance of biodegradable composites. Based on the reasons above mentioned, I recommend publication in Materials.

Minor Comments:

1. In order to extend this study on the degradation of  in the composites, I would suggest to be investigated in more detail. The authors can provide additional evidence using gel permeation chromatography. .

2. In the material characterization part, although the authors have performed SEM and DSC analyses (Figures 3 and 5), the structural features of these composites determined by wide-angle X-ray diffraction should be added. In addition, the weight fractions of PHB and RHF in the composite structure should be discussed in the main text.

3. There is an important paper by Clement Matthew Chan and coworkers entitled “Mechanical properties of poly(3hydroxybutyrateco3hydroxyvalerate)/wood flour composites: Effect of interface modifiers" that should have been discussed and referenced in the revised manuscript (Journal of Applied Polymer Science, 2018, DOI: 10.1002/app.46828).

Author Response

1. In order to extend this study on the degradation of  in the composites, I would suggest to be investigated in more detail. The authors can provide additional evidence using gel permeation chromatography. 

The biodegradability of PHB and its green composites is already well studied in the biopolymer literature and authos consider that this experience is out of the scope of the present study.

2. In the material characterization part, although the authors have performed SEM and DSC analyses (Figures 3 and 5), the structural features of these composites determined by wide-angle X-ray diffraction should be added. In addition, the weight fractions of PHB and RHF in the composite structure should be discussed in the main text.

Commercial PHB and PHBV have been fully characterized by WAXD in some of our previous studies (please see for instance http://dx.doi.org/10.1016/j.polymdegradstab.2016.03.039). However, we have also observed during other previous study from our research group (see https://doi.org/10.1002/app.42633) that the presence of the organic remnants of cell debris or fatty acids from the bioproduction process of PHBV leads to misleading information during WAXD analysis and, thus, this technique becomes not completely useful for crystallinity characterization.

3. There is an important paper by Clement Matthew Chan and coworkers entitled “Mechanical properties of poly(3‐hydroxybutyrate‐co‐3‐hydroxyvalerate)/wood flour composites: Effect of interface modifiers" that should have been discussed and referenced in the revised manuscript (Journal of Applied Polymer Science, 2018, DOI: 10.1002/app.46828).

This previous study has been added to the manuscript as reference 48 and it is also described in lines 91-93.

Reviewer 2 Report

Abstract –

Introduction-

 PHB is a form of PHA’s – biodegradable and bio-based – can you comment on differences between biodegrading properties of regular PP’s or PET’s vs PHB’s (in terms of time or conditions)

Line -47, heavy investments – why it is expensive, reasons?

Are there any established parameters that authors want to attain when they desired to attain highly sustainable composites films – any reference materials in literature or market they benchmark with?

Materials-

 for phr calculations, TGIC and DCP weight or volume is taken into account?

Table 1 shows – TGIC and DCP composition is constant at 1 and 0.25 phr respectively, is there any specific reason for keeping them constant?

The crosslinking and adhesion properties might be altered by varying the TGIC and DCP ratios?

Does a combination of PHB/PHBV5/RHF studied as a control like PHB/RHF (sample 1) because it will give information on the role of PHBV5

Equation 2 needs to be rewritten  O = A500 .L ( I guess L is not a subscript)

Results-

The SEM images do not show the conditions the image is taken, what is the KeV? working distance, etc.

The distribution profile generated cannot be justified with the nature of the sample – the sample consists of different shapes, it’s a heterogeneous sample and especially a does not show any clear distribution of particles? Wondering how authors have come with the histogram?

Please add the sample information that was used for the microscopy in figure 2 title

In table 2- units for T and O?

There is a contradictory statement in lines 401 “the reduction in thermal stability….? Followed by line 407 “the improvement achieved in the thermal stability…” can authors justify?

Author Response

PHB is a form of PHA’s – biodegradable and bio-based – can you comment on differences between biodegrading properties of regular PP’s or PET’s vs PHB’s (in terms of time or conditions)

The concept of biodegradability in terms of time/conditions has been described in more detail in lines 38-40.

 Line -47, heavy investments – why it is expensive, reasons?

This statement has been fully explained and also quantified in lines 48-51.

 Are there any established parameters that authors want to attain when they desired to attain highly sustainable composites films – any reference materials in literature or market they benchmark with?

The main objective is to increase the performance, mainly thermal stability and mechanical ductility, of green composites based on commercial PHB and rice husk containing different amounts of purified sustainable PHBV to make more similar to petrochemical polymers. This has been described in the first part of theIntroduction and particularly in the last paragraph, please see lines 113-116.

 Materials-

  for phr calculations, TGIC and DCP weight or volume is taken into account?

In the plastic industry, the term “phr” is habitually based on weight values since most raw materials are added in solid form.

 Table 1 shows – TGIC and DCP composition is constant at 1 and 0.25 phr respectively, is there any specific reason for keeping them constant?

The TGIC and DCP contents are constant based on the previous works described in the whole paragraph of lines 94-112.

 The crosslinking and adhesion properties might be altered by varying the TGIC and DCP ratios?

Yes, indeed these properties will vary, but these ratios were already analyzed in the previous studies and only the optimal values were selected.

 Does a combination of PHB/PHBV5/RHF studied as a control like PHB/RHF (sample 1) because it will give information on the role of PHBV5

The role of PHBV in the PHB/RHF composite was analyzed by varying the content of the biosustainable copolyester. The number in the sample name refers to the weight content of PHBV in the film.

 Equation 2 needs to be rewritten  O = A500 .L ( I guess L is not a subscript)

This typo was corrected.

 Results-

 The SEM images do not show the conditions the image is taken, what is the KeV? working distance, etc.

This information was fully provided in the experimental part. Please see lines 177-178.

 The distribution profile generated cannot be justified with the nature of the sample – the sample consists of different shapes, it’s a heterogeneous sample and especially a does not show any clear distribution of particles? Wondering how authors have come with the histogram?

Based on the reviewer comment, we have re-analyzed the particle size distribution by dynamic light scattering (DLS) using a laser diffraction analyzer. We have also replaced the low-magnification image in Figure 2a to provide a better overview of the whole particle morphology. This new information has been added in the experimental part, please see lines 180-183, and also in the first paragraph of section 3.1.

 Please add the sample information that was used for the microscopy in figure 2 title

This information was fully described in the Experimental part, please see section 2.4.1.

 In table 2- units for T and O?

These values are dimensionless. Please see for instance, the studies carried out by Sahraee et al. (http://dx.doi.org/10.1016/j.lwt.2016.10.028) or Wang et al. (http://dx.doi.org/10.1016/j.ijbiomac.2016.10.014) in which transparency was determined.

 There is a contradictory statement in lines 401 “the reduction in thermal stability….? Followed by line 407 “the improvement achieved in the thermal stability…” can authors justify?

This is due to the fact that the first sentence relates to the effect of RHF on PHB (thermal stability reduction) whereas the second one relates to the effect of PHBV on the PHB/RHF composite (thermal stability increase)

Round  2

Reviewer 1 Report

Although this study has its own specific objectives and interpretations, the revised version of the manuscript has very slightly changed with providing some explanations and thus may be suitable for publication in its current form.